# Leveraging Machine Learning and Threshold-Free Cluster Enhancement to Unravel Perception of Emotion and Implied Movement

Shyamal Y. Dharia
*Applied Computer Science*
University of Winnipeg
Winnipeg, Canada
dharia-s@webmail.uwinnipeg.ca

Mahdis Hojjati
*Applied Computer Science*
University of Winnipeg
Winnipeg, Canada
hojjati-m@webmail.uwinnipeg.ca

Sergio G. Camorlinga
*Applied Computer Science*
University of Winnipeg
Winnipeg, Canada
s.camorlinga@uwinnipeg.ca

Stephen D. Smith
*Psychology*
University of Winnipeg
Winnipeg, Canada
s.smith@uwinnipeg.ca

Amy S. Desroches
*Psychology*
University of Winnipeg
Winnipeg, Canada
a.desroches@uwinnipeg.ca

*Abstract*— **Understanding the neural mechanisms underlying emotional processing is critical for advancements in emotional neuroscience. This study explores the relationship between emotion and motion perception using Event-Related Potentials (ERPs) in a structured experimental setup. We incorporate subjective intensity ratings to enrich the data by capturing the subjective experiences of participants in response to emotional stimuli with implied motion and no motion. Thirty university students participated in the study, where EEG data was collected and analyzed using threshold-free cluster enhancement (TFCE) for time-domain analysis and Repeated Measures ANOVA for frequency-domain analysis. Furthermore, we developed a multimodal deep learning model to predict subjective intensity levels from EEG-derived features. This model leverages statistical, spectral, and autocovariance features, integrated through a transformer encoder layer, to enhance predictive capability. Our findings contribute to a deeper understanding of emotional processing in the brain and highlight the importance of incorporating subjective measures in neuroscience research.**

*Keywords*— *Electroencephalography (EEG), Event-Related Potentials (ERPs), Implied Motion, Multimodal Deep Learning.*

## I. INTRODUCTION

Brain-Computer interface (BCI) technologies, such as neurosurgical robots inserting thousands of electrodes into the brain [1] and lacing the inside of the blood vessel with electrodes that can record neural activities [2], are advancing. When these systems are connected, it allows researchers to study the neural patterns associated with various studies that are being designed for a specific understanding of neurological conditions. These invasive methods, however, pose risks limiting their academic and clinical use. Consequently, non-invasive methods such as electroencephalography (EEG) are popular for studying neural activities, particularly for understanding emotions [3], [4], [5], [6].

Importantly, emotions are more than the simple perception of facial expressions. Emotions are subjective feelings, but they are experiences that involve complex biological and cognitive [7], [8]. Previous two studies have shown a complex relationship between perception of emotion and motion using Functional magnetic resonance imaging (fMRI) [6] and EEG [5]. The fMRI research revealed that both motion and no-motion stimuli elicit distinct patterns of neural activation, with emotional stimuli enhancing visual processing and motion stimuli increasing activation in multimodal integration areas [6]. Where, "Motion stimuli" include elements of movement, while "no-motion stimuli" do not depict any movement. In contrast, EEG findings demonstrate that these stimuli affect the brain's processing over time, with motion and no-motion stimuli influencing different components like the N200 and LPP, suggesting unique temporal dynamics in neural processing [5]. Although, these two studies showed the importance of controlling implied movement when developing stimulus sets for emotion neuroscience research, they do not incorporate the subjective experiences of participants in response to these stimuli. Incorporating the subjective dimension of emotional stimuli could unveil neural patterns tied to varying emotional intensity levels, adding a significant layer of depth to these investigations.

Furthermore, the recent developments in deep learning for emotion recognition using EEG have been promising. For instance, Song, et al, [9] proposed a general-purpose EEG architecture, called Conformer, for classification task on varied domain-specific datasets, including emotion classification, which takes raw EEG signal to learn global dependencies in the temporal domain and map the learned global dependencies on a topography to locate key information. Further [10], [11], [12], showed the effectiveness of fusing eye movement features with EEG to enhance the predictive capabilities of their AI models for emotion recognition. Inspired by their work, we aim to develop a model for an Event-Related Potentials (ERP) study, a neuroimaging tool with millisecond-level temporal resolution, with integration of subjective responses to emotional stimuli to predict the intensity levels reported by subjects using a multimodal approach. In this framework,

This work was supported by Misericordia Health Center, Mitacs Accelerate (IT39756), and NSERC Discovery Grants (RGPIN-2023-03443 and 418650-2012-RGPIN). Corresponding author(s): Sergio Camorlinga, Stephen Smith, and Amy Desroches.

features are grouped according to different extraction methods to enhance the predictive capability of the model.

The overall contributions of our study are summarized as follows:

1) Our study advances the understanding of affective neuroscience by integrating subjective intensity assessments with implied motion and no motion.

2) We employ statistical approaches, using threshold-free cluster enhancement (TFCE) for time-domain analysis and Repeated Measure ANOVA for frequency-domain analysis. These techniques enable precise differentiation of neural responses, facilitating a deeper understanding of the differences between implied motion with intensities, no motion with intensities, and intensities alone.

3) Our research introduces a multimodal deep learning model that incorporates multiple EEG-derived features for emotional intensity classification. We also provided various important factors that may help experiment design for future emotion studies, especially when subjective responses are required.

## II. RELATED WORKS

Much research has studied the perception of movement and the perception of emotion with different methodologies. Transcranial magnetic stimulation studies have shown that emotions encompass motoric elements. For instance, motor-evoked potentials—small movements in hand muscles following magnetic pulses to the primary motor cortex—are larger when participants are viewing emotional stimuli [13], [14], [15], [16], [17], [18]. In contrast, fMRI, a powerful tool for observing neural activity across the entire brain, has been instrumental in studies that examine how emotions can influence motor regions like the supplementary motor cortex and midcingulate gyrus, suggesting that our emotional responses to stimuli might interact with our movement systems [19], [20], [21] Additionally, another fMRI study highlighted that while the perception of emotion and implied movement activates several brain regions, specific areas in the medial prefrontal and parietal regions are particularly sensitive to both types of information [6]. This suggests a complex interplay between how we process emotions and movement, with certain brain areas playing key roles in integrating these functions. However, due to the relatively slow temporal resolution of fMRI, it was necessary to investigate emotion-movement interactions with additional neuroimaging techniques.

Recent studies have employed ERPs, focusing on N200, P300, and Late Positive Potential (LPP) waveforms. N200 waveform, peaking at 240ms in response to emotional stimuli, is associated with early perceptual processing of sensory stimuli [22], P300 is a positive ERP component which shows maximum amplitude along midline parietal ERP sites at 300-500ms [23] and LPP is associated to responding to intense emotional stimuli [24]. The findings from ERP studies underscore that the brain processes emotion and movement, with significant N200 responses observed across multiple brain regions when participants are exposed to both types of stimuli [5]. Additionally, LPP responses indicate distinct neural patterns to emotional and movement-related information, with significant interactions observed particularly at parietal sites during the first 1000 milliseconds of stimulus encoding. These insights, together with earlier fMRI findings using the same stimuli [6], offer a comprehensive view of how our brains perceive and integrate emotional and movement-related stimuli over time. However, given the subjective nature of emotional perception, where individual responses can significantly vary, the previous ERP study lacks the subjective element. Our research emphasizes the critical need to incorporate subjective emotional experiences. This approach not only enriches our understanding of neural processes but also enhances the applicability and relevance of neuroscience findings from a subjective standpoint.

Furthermore, the datasets used for emotion recognition such as SEED-IV [12] and SEED-V [11] utilize movie clips as stimuli in their EEG-based studies. In contrast, stimuli in ERP studies are brief but effective in eliciting distinct neural responses. While the subjectivity of emotional responses to these stimuli may vary among individuals, the primary objective of ERP studies is to produce an evoked response for each condition, which represents the average waveform across subjects for specific conditions. By incorporating subjective responses collected after each stimulus, our ERP study extends the analysis beyond identifying condition-specific evoked responses, as typically explored in previous research [5].

The SEED datasets, a widely recognized EEG-based emotion recognition datasets, instructs participants to rate their emotional response to each movie clip on a scale from 0 to 5, where 5 indicates the strongest emotional induction and 0 the weakest. For instance, participants are expected to score between 4 and 5 if they experience joy from a joyful video, and 0 if they feel indifferent or if their emotional response is inconsistent. Despite the availability of these comprehensive self-assessments, prior research [9], [10], [11], [12] involving the development of AI models for emotion recognition has primarily focused on the actual stimuli presented, rather than incorporating subjective labels that reflect individual emotional responses. While this approach is valuable, it overlooks the inherently subjective nature of emotions. By training models on genuine subjective responses, we could significantly enhance their utility, enabling them to provide more accurate assessments of an individual's emotional states. This shift towards incorporating subjective experiences into model training also makes them more relevant for finding subjective neural patterns.

## III. METHODS

Overall, this section delineates the diverse methodologies employed in this study, including descriptions of the participants pool, the experimental design, and the stimuli utilized for data collection. For statistical analysis, we detail the parameters for ERP recording and preprocessing, employ TFCE permutation testing, and utilize Repeated Measures ANOVA. Additionally, we outline our approaches for machine

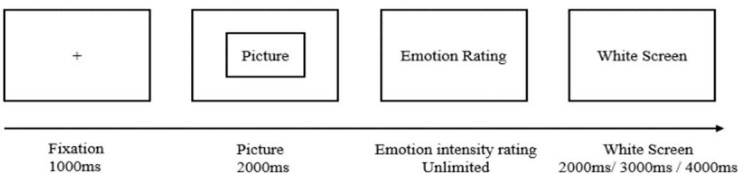

Fig. 1. Initially, participants focus on a fixation point for 1000 ms to ensure attentional readiness. Following this, a picture stimulus is presented for 2000 ms, during which emotional and cognitive responses are elicited. Immediately after the stimulus presentation, participants rate their emotional intensity on a scale from 1 (low) to 4 (high). Each experimental block consists of 60 trials, with participants completing a total of 180 trials throughout the study.

learning, focusing on feature extraction techniques and the design of our network architecture.

Considering emotions are subjective feelings, we added a subjective response (intensity levels) component to our EEG study with a similar set of stimuli from previous studies with implied motion and no motion [5], [6]. As shown in Fig 1, each trial in our study begins with a fixation point displayed for 1000ms. Following this, the stimulus is presented for 2000ms, after which the participant provides an emotional intensity rating, ranging from 1 (low) to 4 (high). This integration of subjective responses is crucial as it allows us to capture the emotional intensities individuals experience in response to different stimuli, enriching our understanding of the subjective dimensions of emotion that objective measures alone may overlook. Finally, to mitigate fatigue, participants had the option to rest after each of the three experimental blocks, each consisting of 60 trials and to reduce learning effects, the presentation order of the stimuli was randomized.

In our ERP study, the comparison conditions are categorized into two primary groups: neutral stimuli and emotional stimuli, each further subdivided based on the presence of implied motion or no motion. Each of these four categories has four levels of intensity which are the subjective responses from participants. For the statistical analysis of these conditions, we employed TFCE [25], a method that facilitates the determination of statistical significance between conditions across all time points without relying on predefined thresholds to identify significant channels or time points. In this study, TFCE was specifically applied to time points to rigorously assess temporal dynamics. For the frequency domain analysis, we utilized the Repeated Measures ANOVA test, a standard approach for evaluating statistical significance between conditions. We specifically used this test to determine the significant differences between raw subjective intensities without considering the underlying motion component. This combination of methods ensures a comprehensive and robust statistical framework.

For our deep learning model, we use 4 feature vectors each of which constitutes a different modality to the network. For instance, statistical features like mean, standard deviation, and variance for each EEG channel are grouped as one modality; Power Spectral Density (PSD) forms another modality; autocovariance another; and differential entropy (DE) features constitute yet another modality. Each feature is stored in an independent feature vector and learned independently through fully connected layers and finally, a transformer encoder layer is utilized to form a communication phase between each feature vector before the classification layer.

### A. Participants

Thirty participants (23 female and 7 males, age; $M = 21.1$, $SD = 4.7$), university students, with normal or corrected-to-normal vision were recruited. Participants received course credits for their participation. All participants provided informed, written consent prior to the beginning of the experiment. All of the procedures in this experiment were performed following the ethical standards of the institutional research ethics board, the National Research Committee ethics requirements, and the 1964 Helsinki Declaration and its later amendments.

### B. Stimuli, Experiment Design and Procedure

The stimuli utilized in this study consisted of 180 colour photographs, 90 negative and 90 neutral stimuli, identical to those employed in previous research [5], [6], where emotional stimuli were limited to negative scenes and the movement within stimuli for both neutral and emotional stimuli was limited to the upper limbs (e.g., needle passed through the thumb). This consistency allows for a direct comparison of results across different studies. The key distinction in our approach lies in the collection of subjective intensity ratings for each stimulus.

Each participant's head was measured in centimetres according to standard practice (i.e., using the nasion and inion as landmarks). Participants were then fitted with a 31-channel EEG cap that best fit their head measurement. After being fitted with the EEG cap, participants were seated in a Whisper Room sound-attenuating booth (Whisper Room Inc., Knoxville, Tennessee). Before starting the experiment, participants received detailed instructions about the experiment's design. Upon completion of these preparatory steps, the experiment was initiated. This structured setup ensures that all participants understand the procedures and contributes to the reliability and consistency of the data collected.

### C. ERP recording & Preprocessing Parameters

Participants were fitted with a 31 Ag/AgCl electrode Brain Products© EasyCAP (actiCHamp Plus, Brain Products GmbH, Gilching, Germany). Data were recorded using a Brain Vision Recorder and amplified via an actiCHamp amplifier at a 500 Hz sampling rate, using Cz as a digital reference. Impedance values were kept below 20 kΩ, the acceptable level as per the manufacturer's guidelines.

Initially, raw EEG data files were loaded for each participant. A high-pass filter of 0.01 Hz was applied to eliminate low-frequency drifts, followed by a notch filter at 60 Hz to remove power line noise. Similar to previous research [5], the data were then re-referenced to the average of the TP9 and TP10 electrodes. Epochs were segmented from the continuous data, spanning from -100 ms to 1000 ms relative to stimulus onset.

An automated artifact rejection procedure using AutoReject [26] was applied to detect and interpolate bad segments within the epochs. Additionally, an Independent Component Analysis (ICA) was performed to identify and remove components corresponding to ocular artifacts. For removing the ICA component in this automated preprocessing pipeline, we utilized Fp1 and Fp2 channels, located near the eyes to create EOG epochs for detecting eye blinks. The ICA was then used to isolate components corresponding to eye movements by evaluating their correlation against the EOG epochs created from the Fp1 and Fp2 channels. Components with a high correlation, indicative of eye movement artifacts, were identified using a maximum absolute score across the EOG channels, with a threshold set at 0.5. Components exceeding this threshold were considered significant for ocular noise and were subsequently excluded, resulting in ICA-cleaned epochs. Following the ICA cleaning, an additional round of automated artifact rejection was applied, and the data was baseline corrected using the pre-stimulus interval from -100 ms to 0 ms. Finally, epochs containing more than ±100 μV are rejected from the data. This preprocessing pipeline allowed us to repair and preserve 4240 out of 5400 epochs, which is ~78%.

### D. Threshold-Free Cluster Enhancement

In our study, the TFCE method was used for our statistical analysis. This method was originally introduced by Smith and Nichols, 2009 [25], and addressed the common issues of smoothing, threshold dependence, and localization in cluster inference within neuroimaging studies. We utilized TFCE because it allows us to identify differences over time, enabling us to differentiate when exactly the significant difference arises between conditions.

To test the null hypothesis at the population level, we drew a sample of subjects from the target population. The initial step involved averaging all trials within each subject across various conditions to obtain subject-specific evoked responses. We then computed the difference between two comparison conditions to generate a condition-specific difference in evoked responses. Finally, we applied a spatial-temporal permutation cluster test, a non-parametric, cluster-level paired t-test tailored for spatio-temporal data [27]. The number of permutations per comparison condition was set to 50,000. We focused on following brain regions: The Frontal region with channels F3, F4, FC1, and FC2; the Central region with C3, C4, and Cz; the Parietal region including P3, P4, Pz, CP1, CP2, CP5, and CP6; the Occipital region comprising O1, O2, and Oz; and the Temporal region consisting of T7, T8, P7, P8, FT9, and FT10.

### E. Feature Extraction and Preprocessing for Machine learning

Clean epochs from Independent Component Analysis (ICA)

and Auto Reject are utilized for feature extraction, aiming to enable machine learning models to predict subjective intensities of emotional stimuli. We employed the following features: Statistical features, including mean, standard deviation, variance, skewness, and kurtosis, are computed for 27 of the total 31 channels. We specifically exclude channels Fp1, Fp2, TP9, and TP10, as they serve distinct purposes in preprocessing: Fp1 and Fp2 are used as EOG channels for ocular blink detection, while TP9 and TP10 serve as offline references. Band power calculations are confined to the Alpha (8-12 Hz) and Beta (12-30 Hz) frequency bands only. These bands are essential for the recognition of emotional stimuli intensities and are analyzed regardless of the stimuli's motion or no motion context, which will be further discussed in the results sections.

Additionally, autocovariance features are extracted for each channel to quantify the linear dependency of the EEG channel over time. The autocovariance for a channel $c$ at the lag $k$ is defined as:

$$AutoCov\,(k) = \frac{1}{T-k} \sum_{t=1}^{T-k} (x_{c,t+k} - \bar{x}_c)(x_{c,t} - \bar{x}_c) \qquad (1)$$

Here, $T$ is the total number of time points, $x_{c,t}$ is the signal value at time point $t$ and $\bar{x}_c$ is the mean of the signal over the time series for channel $c$.

Finally, we calculate Differential Entropy (DE) features, which are extensively used in EEG-based emotion recognition research due to their proven effectiveness [12] Differential Entropy quantifies the uncertainty or complexity in the EEG signal within specific frequency bands. For a fixed segment length of EEG data, DE is computed as the logarithm of the energy spectrum in these bands [12]. The mathematical expression for DE is given by:

$$h(\mathbf{X}) = -\int_{-\infty}^{\infty} \frac{1}{\sqrt{2\pi\sigma^2}} e^{-\frac{(x-\mu)^2}{2\sigma^2}} \log\left(\frac{1}{\sqrt{2\pi\sigma^2}} e^{-\frac{(x-\mu)^2}{2\sigma^2}}\right) dx$$

$$= \frac{1}{2}\log(2\pi e\sigma^2) \qquad (2)$$

Finally, the features are scaled with MinMax standardization and are normalized independently for each feature vector as

$$X_{norm} = \frac{X - X_{min}}{X_{max} - X_{min}} \qquad (3)$$

Where $X$ is the original value, $X_{min}$ and $X_{max}$ are the minimum and maximum values found in the dataset. This scaling adjusts the feature values to a common scale of 0 to 1.

We chose these features because our statistical analysis revealed distinct patterns in both the frequency and time domains of our data. Spectral features from the Alpha and Beta bands are linked to emotional intensities, while statistical features like mean and variance describe signal variability. Autocovariance measures temporal dependencies, and Differential Entropy quantifies signal complexity. These features enable our model to learn various signal characteristics and identify relevant patterns.

## F. Network Architecture

We designed a multi-modal network architecture to effectively capture and combine independent feature representations from four different feature vectors. The architecture includes several stages:

*1) Linear Blocks for Feature Extraction:* Each of the four feature vectors is processed independently through a sequence of fully connected layers, each followed by a non-linear activation function (ReLU) and a dropout layer with a dropout rate of 0.2. This structure allows the network to learn essential features from each vector while reducing their dimensionality. The hidden units in these linear blocks are configured as follows: 128, 32, and 12. Following the fully connected layers, the four processed feature vectors are concatenated. This step stacks the feature vectors along a new dimension, preparing them for subsequent processing by the transformer encoder layers. The resulting tensor has the shape $(b, c, p)$, where $b$ represents the batch size, $c$ represents the number of modalities, and $p$ is the dimension of the features.

*2) Transformer Encoder Layer:* The concatenated features are then fed into a transformer encoder layer, similar to the architecture proposed in [28] for machine translation tasks. This approach in our study is specifically employed to capture similarities and correlations between the different modalities, thereby enhancing the overall representation of the features. The core mechanism enhancing the features within the transformer encoder is the self-attention block, which is mathematically represented as:

$$Attention(Q, K, V) = Softmax\left(\frac{Q\,K^T}{\sqrt{d_k}}\right)V \qquad (4)$$

Where $Q$ (queries), $K$ (keys), and $V$ (values) are the input features derived by an independent linear layer, making them learnable parameters for our model. $d_k$ is the dimension of the keys. The subsequent layers of the transformer encoder are similar to [9], [28].

Finally, the output from the transformer encoder layers is averaged across the second dimension $(b, c, p)$, resulting in a mean-pooled vector that effectively summarizes the learned representation into a single vector per sample. This vector serves as the input to the classifier block, which consists of a dropout layer with a dropout rate of 0.5, followed by a linear layer for classification output.

## IV. EXPERIMENTAL AND RESULTS

In this section, we conduct analyses to investigate differences between implied motion and no motion stimuli for both neural and emotional conditions using TFCE permutation testing. Additionally, we use repeated measures ANOVA tests to identify relevant evidence for applying machine learning to predict raw intensity ratings. Finally, we evaluate the effectiveness of our multi-modal model for prediction.

### A. Threshold-Free Cluster Enhancement permutation testing

We investigated the neural responses to implied motion versus no motion across varied subjective intensities. Our approach focused on exploring how the subjective intensities of the stimuli influence neural patterns, providing insights into the subjective experience of motion. To identify significant differences in neural activity between conditions, eight stimulus types were used in this study, categorized into four levels of motion intensity (Motion 1, 2, 3, 4) and four levels of no motion intensity (No Motion 1, 2, 3, 4), with intensities 3 and 4 combined due to a smaller number of trials. We employed TFCE permutation testing to analyze the data. As illustrated in Figures 2 and 3, the TFCE results reveal the spatial and temporal dynamics of brain activity differentiating between implied motion and no motion. The colour bar indicates the microvolt differences evoked between the conditions, and the electrode sensors marked as white dots show where significant differences were found. Notably, the P300 and N200 responses did not show significant differences, and thus only the LPP window (approximately 450ms to 1000ms) was considered for analysis. The most consistent effects were noted in the frontal, central, and posterior regions across both neutral and emotional stimuli conditions.

First, we analyzed the neural responses to neutral stimuli with implied motion compared to no motion, focusing on different intensity levels. As shown in Fig 2, we observed notable differences when comparing the lowest motion intensity (Motion 1) to the lower no motion intensities (No Motion 1 and 2). Interestingly, as the intensity of the stimuli increased, the LPP effects decreased. There were no significant differences when Motion 1 was compared to higher no motion intensities (No Motion 3+4), suggesting similar neural responses between these conditions. Further analysis showed that Motion 1 elicited responses comparable to the higher no motion intensities, indicating that even low-intensity motion stimuli can evoke strong neural reactions. However, when comparing Motion 2 across different levels, the differences were more pronounced against No Motion 1 than No Motion 2, and no significant differences were observed when comparing Motion 2 with the combined higher intensities (No Motion 3+4). Additionally, when we analyzed the highest combined intensities of motion and no motion (3+4), we found detectable effects, though they were less pronounced compared to the lower intensities. This suggests a diminishing gradient of neural reactivity with increasing stimulus intensity, possibly indicating an upper limit in how the sensory and emotional response systems process these stimuli.

In the case of emotional stimuli, our results revealed more consistently significant effects compared to neutral stimuli. For instance, as shown in Fig 3, significant neural responses were observed when comparing the lowest intensity of motion (Motion 1) against the lower no motion intensities (No Motion 1 and 2), and these effects remained strong even with increased motion intensity. Notably, no significant differences were detected when Motion 1 was compared to the higher no motion intensities (No Motion 3+4). The trend continued with Motion 2, where effects against No Motion 1 and 2 were consistent, but

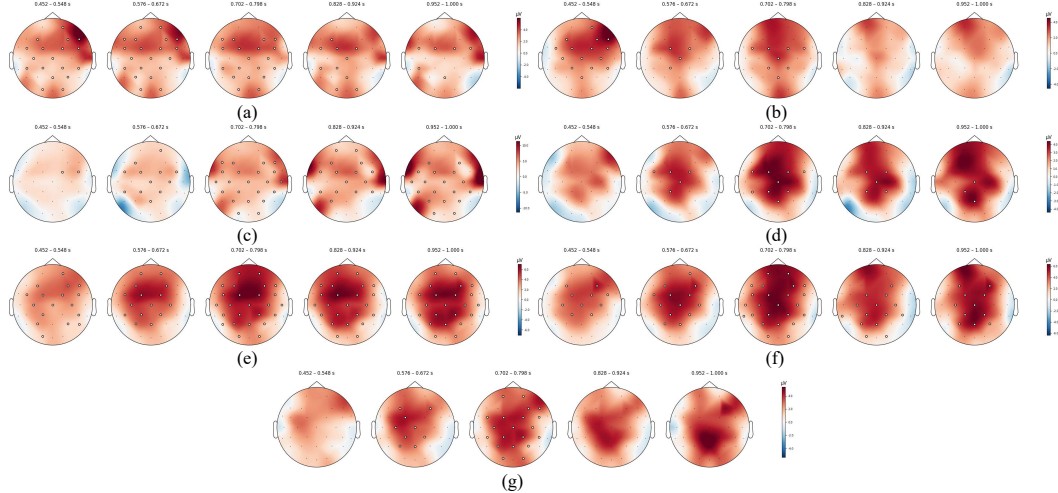

Fig. 2 Results of neutral stimuli motion Vs No motion conditions TFCE mapped on topography. (a) Motion Intensity 1 Vs No Motion Intensity 1, (b) Motion Intensity 1 Vs No Motion Intensity 2, (c) Motion Intensity 2 Vs No Motion Intensity 1, (d) Motion Intensity 2 Vs No Motion Intensity 2, (e) Motion Intensity 3+4 Vs No Motion Intensity 1, (f) Motion Intensity 3+4 Vs No Motion Intensity 2, and (g) Motion Intensity 3+4 Vs No Motion Intensity 3+4

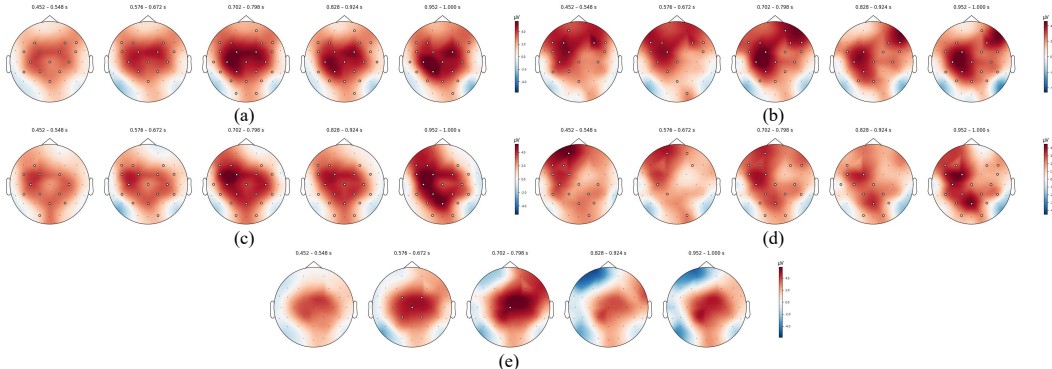

Fig. 3 Results of emotional stimuli motion Vs No motion conditions TFCE mapped on topography. (a) Motion Intensity 1 Vs No Motion Intensity 1, (b) Motion Intensity 1 Vs No Motion Intensity 2, (c) Motion Intensity 2 Vs No Motion Intensity 1, (d) Motion Intensity 2 Vs No Motion Intensity 2, and (e) Motion Intensity 3+4 Vs No Motion Intensity 1

diminished when compared to the combined higher no motion intensities (No Motion 3+4). Most intriguingly, at the highest motion intensities (3+4), only marginal effects were noted against No Motion 1 between 570ms and 900ms, after which these effects dissipated. No effects were found against other no-motion intensities at these high levels of motion. This finding is particularly significant as it may suggest that at high intensities, the differential effects of motion versus no motion begin to vanish. Such a phenomenon could imply that as stimuli become more emotionally intense, the emotional content of the stimuli becomes more important than information related to movement.

*B. Repeated ANOVA test*

The results of permutation testing revealed significant differences between motion and no-motion conditions. To further investigate, we examined the differences in raw intensities without considering the implied motion and no motion stimuli conditions. We conducted a Repeated Measures ANOVA test. The features used for this test were PSD values from the Alpha (8-12 Hz) and Beta (12-30 Hz) frequency bands.

The Bonferroni corrected post hoc comparisons for the Alpha band (8-12 Hz) with neutral stimuli showed significant differences in raw intensities between Level 1 and Level 4 $(t(3) = -3.903, p = 0.002)$, as well as a marginal difference between Level 2 and Level 4 $(t(3) = -2.721, p = 0.052)$. No significant differences were found between Level 3 and Level 4 $(t(3) = -1.879, p = 0.394)$. A similar trend was observed for emotional stimuli, where Level 1 and Level 4 $(t(3) = -4.506, p < 0.001)$ and Level 2 and Level 4 $(t(3) = -4.142, p < 0.001)$, shows significant difference but no significant differences were found between Level 3 and Level 4 $(t(3) = -2.358, p = 0.131)$.

Furthermore, for the Beta band with neutral stimuli, a similar trend is observed. Significant differences were found between Level 1 and Level 4 $(t(3) = -3.860, p = 0.002)$ and between Level 2 and Level 4 $(t(3) = -2.317, p = 0.041)$, while no significant differences were observed between Level 3 and Level 4 $(t(3) = -2.137, p = 0.223)$. For the Beta band with emotional stimuli, significant differences were observed between Level 1 and Level 4 $(t(3) = -5.156, p < 0.001)$ and between Level 2 and Level 4 $(t(3) = -4.595, p < 0.001)$. Additionally, a significant difference was found between Level 1 and Level 3 $(t(3) =$

$-2.677, p = 0.058)$, though it was less pronounced compared to the others. No significant differences were observed between Level 3 and Level 4 ($t(3) = -2.479, p = 0.097$).

The Repeated Measures ANOVA test shows that it's appropriate to group the intensity levels for analysis. Specifically, Levels 1 and 2 can be grouped, and Levels 3 and 4 can be grouped. This grouping decision is based on the finding that Levels 1 and 2 do not show significant differences in PSD values across frequency bands, suggesting they are similar. Although Level 3 does not consistently show significant differences from Level 2, it similarly lacks significant differences when compared to Level 4. Thus, combining Levels 1 and 2 into one class and Levels 3 and 4 into another class simplifies the data structure and supports more effective classification in machine learning by reflecting the statistical relationships found in the data. However, despite these groupings, there remains a class imbalance in the dataset. Finally, Together with TFCE, these statistical methods provided a comprehensive view, showing that both time-domain and frequency-domain analyses are crucial for a deeper understanding of our data. They enabled us to discern subtle but important patterns that might otherwise be missed, thereby enriching our findings.

### C. Machine Learning

For machine learning on our ERP data, we simply decided to use emotional stimuli for classification as emotional stimuli indicated the main effects between intensities regardless of implied motion and no motion.

*1) Training Strategy:* During the training phase, our model employed two distinct subject-dependent strategies: within-subject analysis and classical 3-fold cross-validation on the entire dataset. The within-subject approach is particularly advantageous for leveraging subject-specific features, especially useful when prior data from a subject is available, allowing for more personalized model performance. In contrast, the classical 3-fold cross-validation offers insights into global feature relationships across all participants. It is important to note that in the within-subject analysis, we had to exclude data from some participants who either lacked ratings across all intensity levels or had insufficient samples, even after combining intensity levels 3 and 4.

*2) Training Details:* For training the within-subject model, we allocated 50 percent of the data for training and 50 percent for testing, due to data limitations. We omitted a validation set and limited training to 100 epochs, utilizing the F1 score (macro) as our performance metric. This approach assumes the availability of initial trial subjective responses from subjects for real-time predictions.

In contrast, for the classical 3-fold cross-validation, we partitioned the data into 60 percent for training, 10 percent for validation, and 30 percent for testing. Both training strategies employed a batch size of 32, using the Adam optimizer with a learning rate of 0.0001, and capped training at 100 epochs.

We used PyTorch on Python 3.11 with an NVIDIA RTX A4000 GPU for implementation. Given the class imbalance, the F1 score (macro) was selected as the evaluation metric. Additionally, we adjusted the cross-entropy loss function by setting class weights to address and compensate for the imbalanced class distribution.

*3) Machine learning results:* To fine-tune our model, we initially conducted hyperparameter tuning using data from a single subject. The optimal settings determined were as follows: the number of self-attention heads in the transformer encoder layer was set to 6, a total of 3 transformer encoder layers were stacked, a dropout rate of 0.2 was implemented, and the feed-forward dimension was configured to be twice the input dimension, which is 12. These parameters were then consistently applied across the training phase for all subjects.

Our model achieved an overall F1 score (macro) of 65.2% across all subjects. Employing the same hyperparameters from the within-subject training, we conducted a 3-fold cross-validation on the entire dataset. The model achieved a comparable F1 score (macro) of 63.7%. A significant challenge encountered was the class imbalance, with 90% of samples from intensity levels 1 and 2, and only 10% from levels 3 and 4, which hindered the model from reaching our anticipated performance levels. To address the imbalance, we adjusted the class weight ratios in our loss function to more heavily penalize misclassifications of the minority class. However, this adjustment did not significantly enhance performance. This underscores the complexity of predicting subjective emotional responses using AI. Nevertheless, an F1 score of 65.2% and 63.7% with within-subject and 3-fold cross-validation offers promising indications that enhancing the dataset, particularly by increasing the number of samples from levels 3 and 4 class, could improve model performance.

## V. DISCUSSION

Our study examined neural responses to implied motion versus no motion at varying subjective intensities. The TFCE permutation testing showed that, primarily the difference between implied motion and no motion stimuli is at frontal, central, and posterior brain regions, especially in the LPP time window (~450ms to 1000ms). For neutral emotions at high-intensity level 3+4, effects are distinguishable between motion and no motion stimuli but with less significant effects throughout the LPP window. In contrast, for negative stimuli, significant effects were absent between higher intensity levels (3+4) and lower levels (1 and 2), potentially indicating that at high emotional intensities, the emotion in the stimuli becomes more important to the perceiver than the motoric information.

The Repeated Measures ANOVA test on the other hand for raw intensity levels, without considering the motion component, showed that both Alpha and Beta frequency bands exhibited significant differences between the 1 & 2 Vs 4 intensity levels. Interestingly, intensity level 3 has no significant differences between either low levels or high levels of intensity. This may suggest that different people used the 1-4 scale in different ways; for example, some people might have been reluctant to enter "4" because the images weren't upsetting, so a "3" became the default response for anything that was very emotional. Other people might have followed the

experiment instructions and used the full 1-4 scale. This issue has been reflected in our machine learning model, where our model, yet capable, couldn't learn the complex and overlapping intensities regardless of various domain features. In future studies, first, we plan to introduce more intensely negative emotional stimuli which might prompt subjects to select higher intensity ratings. This could effectively widen the gap between low and high intensity ratings, providing clearer distinctions for analysis. Second, we plan to ask participants to complete some practice trials with the experimenter to calibrate their responses better. So, the experimenter could ask participants to articulate why they selected 1, 2, 3, or 4 for some practice trials so that participants would create a mental framework for how they should use all four values in the actual experiment. We think that if the rating was calibrated like this, then the values being used for the machine learning would be more useful, which in turn, would increase the model's performance.

The clinical applications of our findings could be significant, especially in the context of emotion recognition and neurological assessments. Future studies will aim to investigate neural responses in older adults or individuals with emotion regulation issues. This could lead to the development of brain health assessments, using the baseline data from our study on a young, healthy cohort to identify what is expected in a healthy brain. Identifying similar EEG biomarkers in individuals with emotion regulation difficulties could provide valuable insights for early diagnosis and intervention. Additionally, an AI model that predicts subjective emotional intensities could enhance diagnosis by potentially predicting emotional outbursts.

## VI. CONCLUSION

The primary aim of this study was to show the distinct roles of emotional content and implied motion across varying subjective intensity levels. Our findings indicate that at higher emotional intensities, the effects of motion versus no motion dissipate and the emotion in the stimuli becomes more important to the perceiver than the motoric information. Moreover, the repeated measures ANOVA revealed a significant difference between low-intensity ratings (1 and 2) and high ratings (4). However, intensity level 3 did not show a significant difference compared to other levels, suggesting an overlap at this mid-point which posed challenges for our machine learning model. This study also advances our understanding of stimulus selection in emotion-related ERP studies by highlighting the importance of considering motion as a potential variable. By fine-tuning the control or utilization of motion in stimuli, future research can better isolate and examine the neural correlates of emotional processing.

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
