# OpenReview forum: "Leveraging Machine Learning and Threshold-Free Cluster Enhancement to Unravel Perception of Emotion and Implied Movement"
_IEEE.org/EMBS/BHI/2024/Conference — IEEE BHI'24_

### Official Review · Reviewer_SwKL · 2024-08-06
**Review of "Leveraging Machine Learning and Threshold-Free Cluster Enhancement to Unravel Perception of Emotion and Implied Movement"**

**Overall Rating:** 5
**Confidence:** 2

**Other Quality Metrics:**

(a) Clarity of writing; Poor
(b) Clinical Significance; Good
(c) Methodological Novelty; Fair
(d) Experiments and Results; Fair

**Questions For The Authors:**

What was the rationale for choosing threshold-free cluster enhancement (TFCE) and repeated measures ANOVA for your analysis? How do these methods specifically contribute to your findings?
Why did you choose the specific features and architecture for your multimodal deep learning model? Could you provide more detail on how these choices were made and how they align with the goals of the study?

**Strengths:**

The study addresses an important and timely topic in the field of affective neuroscience by exploring the relationship between emotion and motion perception. The use of subjective intensity ratings alongside EEG data is a novel approach that could provide deeper insights into the neural processing of emotions. The development of a multimodal deep learning model for predicting emotional intensity is an interesting application of machine learning techniques in neuroscience.

**Summary Of The Paper:**

The paper explores the relationship between emotional processing and motion perception using Event-Related Potentials (ERPs). It aims to predict subjective intensity levels from EEG data through the use of a multimodal deep learning model that incorporates various EEG-derived features. The study focuses on understanding the neural mechanisms underlying emotional processing by analyzing data through threshold-free cluster enhancement (TFCE) and repeated measures ANOVA. The authors claim that their findings contribute to affective neuroscience by highlighting the importance of incorporating subjective measures.

**Weaknesses:**

Despite the study’s interesting premise, the paper is written in a confusing manner, making it difficult to follow the authors' goals and arguments. The introduction and background sections are overloaded with information that is not clearly connected to the study's objectives. As a result, the central goal of the research is not well-defined, leaving readers unclear about the primary research question or hypothesis being tested.

The presentation of results lacks sufficient detail, particularly in the description of how the data was analyzed and interpreted. The statistical methods employed, such as TFCE and repeated measures ANOVA, are mentioned, but the rationale for choosing these methods is not adequately explained. Moreover, the results are presented in a way that is difficult to interpret, with key findings not clearly highlighted or discussed in the context of the existing literature.

The motivation for the methods used, particularly the machine learning model, is poorly articulated. The paper does not sufficiently justify why the specific features and model architecture were chosen, nor does it provide a clear explanation of how these methods advance the understanding of emotional processing. The lack of clarity in the methodological section undermines the credibility of the findings and leaves the reader questioning the validity of the approach.

---

### Official Review · Reviewer_8K67 · 2024-08-10
**Leveraging Machine Learning and Threshold-Free Cluster Enhancement to Unravel Perception of Emotion and Implied Movement**

**Overall Rating:** 6
**Confidence:** 3

**Other Quality Metrics:**

a) Good: clarity of writing could be improved
b) Fair: clinical applicability of the work is somewhat limited
c) Great: it seems like the approach of TFCE and including no motion stimulus is novel; however it seems like the approach could have considered methods to better class imbalance more
d) Good: again the approach seems good but not complete so it seems like the results were not as conclusive as expected.

**Questions For The Authors:**

1. Is “no-motion” and “motion” a widely accepted terminology in the context of your study? The concept became clearer only after reading the methods section, and it might benefit from an earlier and more explicit definition.
2. Considering that repeated measures ANOVA was used, is there a way to account for potential changes in how participants score over time, such as fatigue or learning effects?
3. The study mentions challenges with class imbalance in the dataset, yet no methods such as oversampling or SMOTE were employed to address this. Was there a particular reason for not using these techniques? Would you consider these methods in future studies to improve the model’s performance?

**Strengths:**

Incorporation of Subjective Ratings: Including subjective intensity ratings adds depth to the analysis, allowing for a more nuanced understanding of how individuals perceive and process emotional stimuli. This approach enhances the ecological validity of the findings.

Innovative Modeling Approach: The application of a multimodal deep learning model to classify emotional intensity based on EEG features is a sophisticated approach that leverages different types of EEG-derived data to improve classification accuracy.

**Summary Of The Paper:**

The authors present three main contributions: (1) integrating subjective intensity ratings for both no-motion and motion stimuli, (2) using TFCE and repeated measures ANOVA to differentiate neural responses across three datasets, and (3) developing a multimodal deep learning model with EEG-derived features for emotional intensity classification. The study is focused on understanding how motion-related stimuli influence emotional processing at the neural level, emphasizing the role of subjective experiences in shaping these responses.

**Weaknesses:**

Figures: The text and legends in Figures 1 and 2 are difficult to read due to their small size. If the color mapping and timing are consistent across the figures, larger, centralized labels could improve readability. Additionally, the figures would benefit from clear indicators (e.g., asterisks) to quickly convey which comparisons are statistically significant, making the results more accessible.

Terminology Clarification: The terms “no-motion” and “motion” used to describe stimuli are not clearly defined early in the paper, which might cause confusion for readers unfamiliar with this research area. Clarifying these terms at the outset would improve the paper’s clarity.

---

### Official Review · Reviewer_EQhj · 2024-08-13
**Review of Submission 292**

**Overall Rating:** 7
**Confidence:** 3

**Other Quality Metrics:**

Clarity of Writing: Good
Clinical Significance: Fair
Methodological Novelty: Good
Experiments and Results: Good

**Questions For The Authors:**

1. How do you plan to address the class imbalance issue in your machine learning model in future studies?
2. Could you elaborate on the potential clinical applications of your findings, particularly in emotion recognition or neurological assessments?
3. How do you see your findings being generalized to a more diverse population beyond the university student sample?

**Strengths:**

1. Innovative Integration of Subjective Measures: The study effectively incorporates subjective intensity ratings, enriching the data and providing a more comprehensive understanding of emotional processing.
2. Advanced Methodological Approaches: The use of TFCE and Repeated Measures ANOVA for statistical analysis is a strong methodological choice, allowing for precise differentiation of neural responses across conditions.
3. Multimodal Deep Learning Model: The introduction of a multimodal deep learning model that integrates various EEG-derived features is a significant contribution, showcasing the potential for AI to predict subjective emotional states.
4. Clear Experimental Design: The structured experimental setup, including the careful selection of stimuli and detailed preprocessing of EEG data, ensures the reliability and validity of the findings.

**Summary Of The Paper:**

The paper titled "Leveraging Machine Learning and Threshold-Free Cluster Enhancement to Unravel Perception of Emotion and Implied Movement" explores the neural mechanisms underlying emotional processing, focusing on the relationship between emotion and motion perception. Using Event-Related Potentials (ERPs) and electroencephalography (EEG) data, the study investigates how subjective intensity ratings of emotional stimuli with implied motion and no motion influence neural responses. The authors employ advanced statistical techniques, such as threshold-free cluster enhancement (TFCE) for time-domain analysis and Repeated Measures ANOVA for frequency-domain analysis, to differentiate neural activity patterns. Additionally, the study introduces a multimodal deep learning model that integrates EEG-derived features to predict subjective intensity levels, advancing the understanding of affective neuroscience.

**Weaknesses:**

1. Limited Generalizability: The study's findings are based on a relatively small sample size of 30 university students, which may limit the generalizability of the results. Expanding the sample size and diversity of participants would strengthen the study's external validity.
2. Complexity of Methodology: The use of advanced statistical and machine learning methods, while innovative, may pose challenges for replication and understanding by researchers not well-versed in these techniques. Additional explanations or simplifications could make the study more accessible.
3. Class Imbalance in Machine Learning Model: The study notes a class imbalance in the dataset used for training the machine learning model, which may affect the model's performance. Addressing this issue or providing a more detailed discussion of its impact on the results would improve the paper.
4. Limited Discussion of Practical Applications: While the study offers significant theoretical contributions, the practical implications of the findings, especially in clinical or real-world settings, are not fully explored. Expanding on how these insights can be applied in practice would enhance the paper's impact.

---

### Decision · Program_Chairs · 2024-09-23

Accept